# Volunteers Recruitment, Retention, and Performance during the CSMON-LIFE (Citizen Science MONitoring) Project and 3 Years of Follow-Up

Stefano Martellos [1,2,*], Elena Pittao [1], Donatella Cesaroni [3], Alessio Mereu [4], Damiano Petruzzella [5], Manuela Pinzari [3], Valerio Sbordoni [3], Giuliano Tallone [6] and Fabio Attorre [2,7]

1   Department of Life Sciences, University of Trieste, 34127 Trieste, Italy; epittao@units.it
2   Centro Interuniversitario di Ricerca per la Biodiversità Vegetale Big Data—PLANT DATA, Department of Biological, Geological and Environmental Sciences, Alma Mater Studiorum University of Bologna, 40126 Bologna, Italy; fabio.attorre@uniroma1.it
3   Department of Biology, Tor Vergata University of Rome, 00133 Rome, Italy; donatella.cesaroni@uniroma2.it (D.C.); manuela.pinzari@uniroma2.it (M.P.); valerio.sbordoni@uniroma2.it (V.S.)
4   Divulgando S.R.L., 34135 Trieste, Italy; mereu@divulgando.eu
5   CIHEAM Bari, 70010 Valenzano, Italy; petruzzella@iamb.it
6   Agenzia Regionale del Turismo, Regione Lazio, 00185 Rome, Italy; gtallone@regione.lazio.it
7   Department of Environmental Biology, Sapienza University of Rome, 00185 Rome, Italy
*   Correspondence: martelst@units.it; Tel.: +39-040-5583889

**Abstract:** Volunteers' contribution to research is growing, especially since the beginning of the 21st century. Given the constant increase of Citizen Science initiatives, recruiting strategies have to be planned properly. Retention is pivotal as well, especially when time is invested in volunteers' training. However, practically no follow-up data are available on retention after major Citizen Science initiatives. CSMON-LIFE (Citizen Science MONitoring) was a 42-month project (2014–2017) funded by the European Commission in the framework of the LIFE+ programme (LIFE13 ENV/IT/842). It aimed at increasing awareness on Citizen Science among citizens, researchers, and decision makers in Italy. During CSMON-LIFE, recruitment was based on extensive awareness raising actions on different media. In total, 5558 volunteers were engaged in different field activities during the project and its follow-up. They gathered a total of 30062 geo-referenced observations, each with an image of the reported organism. Their activities were organized in campaigns, each devoted to a different topic. This study aims at investigating volunteers' performance and retention in the funded period of CSMON-LIFE (December 2014–November 2017) and in its after-LIFE follow-up period (December 2017–November 2020), for a total of 72 months.

**Keywords:** alien species; biodiversity data; field observation; participation

## 1. Introduction

The definition of Citizen Science is somehow fluid and can include different forms and levels of volunteers' involvement in research activities [1–4]. Citizen Scientists can be defined as "non-scientists who help to analyze or collect data as part of a researcher-led project" [5]. The contribution of volunteers to research activities is far from being new [6]. Probably, the first modern Citizen Science project is the Christmas Bird Count, which has been held since December, 1900 [7]. In the last decades the contribution of volunteers' data—especially in the field of biodiversity—has increased significantly [8]. Some examples are the Global Biodiversity Information Facility (GBIF) [9], in which ca. 55% of records are produced by volunteers [10], and the eBird project at Cornell Lab of Ornithology, which receives ca. 25 million observations per month [5]. A study on data collection of water quality in seven US states demonstrates the major contribution of Citizen Science programs

to ecological databases [11]. Even if non-expert contributions to science have inherent risks related to data quality [12], mostly due to the normally limited training of contributors [13] or the absence of formal scientific methods [14], there is evidence that volunteers can collect useful, high-quality data [15–19].

Citizen Science Data could be relevant for monitoring the achievement of the 17 UN Sustainable Development Goals (SDGs). Even if Citizen Science Data are not currently included in SDG data acquisition, it has been demonstrated [20] that Citizen Science has the potential to contribute to all 17 SDGs, and especially in the environmental domain (SDGs 6, 11, and 15). A study [21] issued by the EU commission highlighted that environmental Citizen Science projects already contribute to a diversity of SDGs, in particular to health and well-being (SDG 3), climate mitigation and adaptation (SDG 13), terrestrial nature conservation (SDG 15), and global partnership for sustainable development (SDG 17). Thus, Citizen Science Data are suggested as a useful resource for complementing SDG reporting [22].

The motivation of volunteers for contributing to Citizen Science projects has been thoroughly discussed [23,24], evidencing that it has a dynamic nature, which evolves in time as the engagement and skills of volunteers progress. The motivation can arise from both self-directed and altruistic motives, and hence Citizen Science projects should meet them in order to ensure satisfaction and a high retention [25]. Plus, given the constant increase in the number of Citizen Science initiatives [8], a certain degree of competition for recruitment is to be expected. Thus, Citizen Science projects which will be the most effective in meeting volunteers' motivations will also achieve higher recruitment and retention rates. Recruiting strategies have to be planned properly, since they influence the composition of the volunteer groups which will participate in the project, and thus the quality and quantity of data they will produce [26]. Plus, a lack of commitment from volunteers could lead to gaps in the data across time and space [13,27]. The retention of volunteers, on the other hand, is one of the most challenging issues in Citizen Science [28]. Retention is more an issue related to participants' satisfaction than motivation. While it has been measured in some studies [29–34], a univocal definition is not provided. Plus, retention is often measured during a project, but no follow-up data (to our knowledge) are available to highlight how many volunteers remain engaged after the end of a project.

This study aims at investigating volunteers' performance (in terms of ratio correct/wrong observations) and retention (expressed in duration of participation) for a total of 72 months from 01 December 2014 to 30 November 2020, in the funded period (36 months, from 01 December 2014 to 30 November 2017) of the LIFE+ project CSMON-LIFE, and its follow-up (after-LIFE) period (36 months, from 01 December 2017 to 30 November 2020). CSMON-LIFE (Citizen Science MONitoring, http://www.csmon-life.eu, accessed on 26 April 2021) was a 42-month project (2014–2017) funded by the European Commission in the framework of the LIFE+ programme (LIFE13 ENV/IT/842), aimed at increasing the awareness on the effectiveness of Citizen Science among citizens, researchers, and decision makers, thus generating a virtuous circle of data gathering, analysis, and usage, for producing novel and more effective environmental policies. Volunteers were engaged in reporting observations collected through an App (iOS and Android). The observations, verified by experts, where then aggregated in the repositories of the National Biodiversity Network [35] of the Italian Ministry of Environment.

The activity of volunteers was organized in campaigns, each devoted to a group of organisms (plants, insects, lichens, etc.) or to a specific goal (contests, "BioBlitzes", etc.). Most of the campaigns were opened to all Italian volunteers, while some were restricted to smaller geographic extents. However, these limitations were not forced, and thus any volunteer could report observations from any location. The CSMON-LIFE platform has also been adopted as an observation-gathering tool by several stakeholders, with the creation of "guest" campaigns such as "Urban Nature", developed in cooperation with the Italian World Wildlife Fund (WWF), and the "City Nature Challenge" contest held in Rome in 2018 ("CNCRome2018").

## 2. Materials and Methods

Data were collected for 36 months during CSMON-LIFE (M 1–36), and during 36 months (M 37–72) of follow-up, officially defined as "after-LIFE" in LIFE projects. Each observation is stored together with a unique ID, species name (or the closest higher taxon), date of collection, digital image, ID of the campaign, and ID of the volunteer. Each observation has a number in the system according to its verification status (0 = not verified; 1 = positively verified; 2 = negatively verified). Volunteers are counted as unique individuals, even if in several cases they are "collective", i.e., unique IDs are adopted by teams of volunteers. This was especially true as far as school classes were concerned during school contests, which involved 234 schools and ca. 5600 students.

Retention was estimated on the basis of the dates of the first and the last observation sent by each volunteer, and measured in months. As an example, if a volunteer sent the first observation in March 2015, and the last in June 2016, the retention was 16 months. Volunteers were divided into 6 retention groups: 1 month (i.e., the first and last observations occurred in the same month, or the volunteer sent one observation only), 2–5, 6–10, 11–20, 21–30, and >30 months.

The volunteers were divided into nine age classes based on their year of birth: 1921–30, 1931–40, 1941–1950, 1951–1960, 1961–1970, 1971–1980, 1981–1990, 1991–2000, and 2001–2008. Given the limited numbers (1, 3, and 20), the first three classes were merged into one (<=1950). The last class (2001–2008) is limited to 8 years, since the youngest participants were engaged in activities with elementary schools during 2016, when they were aged 7–8. The number of volunteers recruited in M 1–36 and retained in M 37–72 was calculated on the basis of the month of recruitment and retention.

The analysis of the retention and age classes was restricted to a subset of 1373 volunteers, i.e., those who provided reliable data at registration and did not cancel their registration during the 72 months of the study. As for the reliability of data at registration, several volunteers input erroneous or ambiguous birth dates. Furthermore, volunteers who cancelled their registration had their data removed from the database, and thus could not be included in the analysis. Basic statistics were performed with StatSoft STATISTICA 6.0 and the package EZR version 1.54 on R commander version 2.7–1.

## 3. Results

A summary of the data for the funded period (M 1–36), the follow-up (M 37–72), and overall is reported in Table 1. Among the 5558 volunteers enrolled in project activities, 1852 reported observations. In M 37–72, 367 of 445 volunteers were new recruits, while 78 were retained from M 1-36 (retention rate: 5.25%). Overall, 30062 observations were collected, of which 9174 in M 37–72 (43.92% of M 1–36). The error rate was lower in M 37–72 than in M 1–36 (7% vs. 21%). The trend of observations evidences a certain seasonality, with most observations collected during spring (Figure 1). The flow of observations increased during the funded period, hitting a maximum in spring 2017, towards the end of M 1–36, and decreased in M 37–72.

**Table 1.** Summary of data.

|  | **M 1–36 Period** | **M 37–72 Period** | **Overall Period** |
|---|---|---|---|
| Duration months | 36 | 36 | 72 |
| Volunteers | 1485 | 445 (78 retained, 367 new) | 1852 |
| Active campaigns | 36 | 30 (23 continuing, 7 new) | 43 |
| Total observations | 20,888 | 9174 | 30,062 |
| Correct observations | 15,836 (76%) | 7691 (84%) | 23,527 (78%) |
| Wrong observations | 4431 (21%) | 654 (7%) | 5085 (17%) |
| Uncertain observations | 621 (3%) | 829 (9%) | 1450 (5%) |

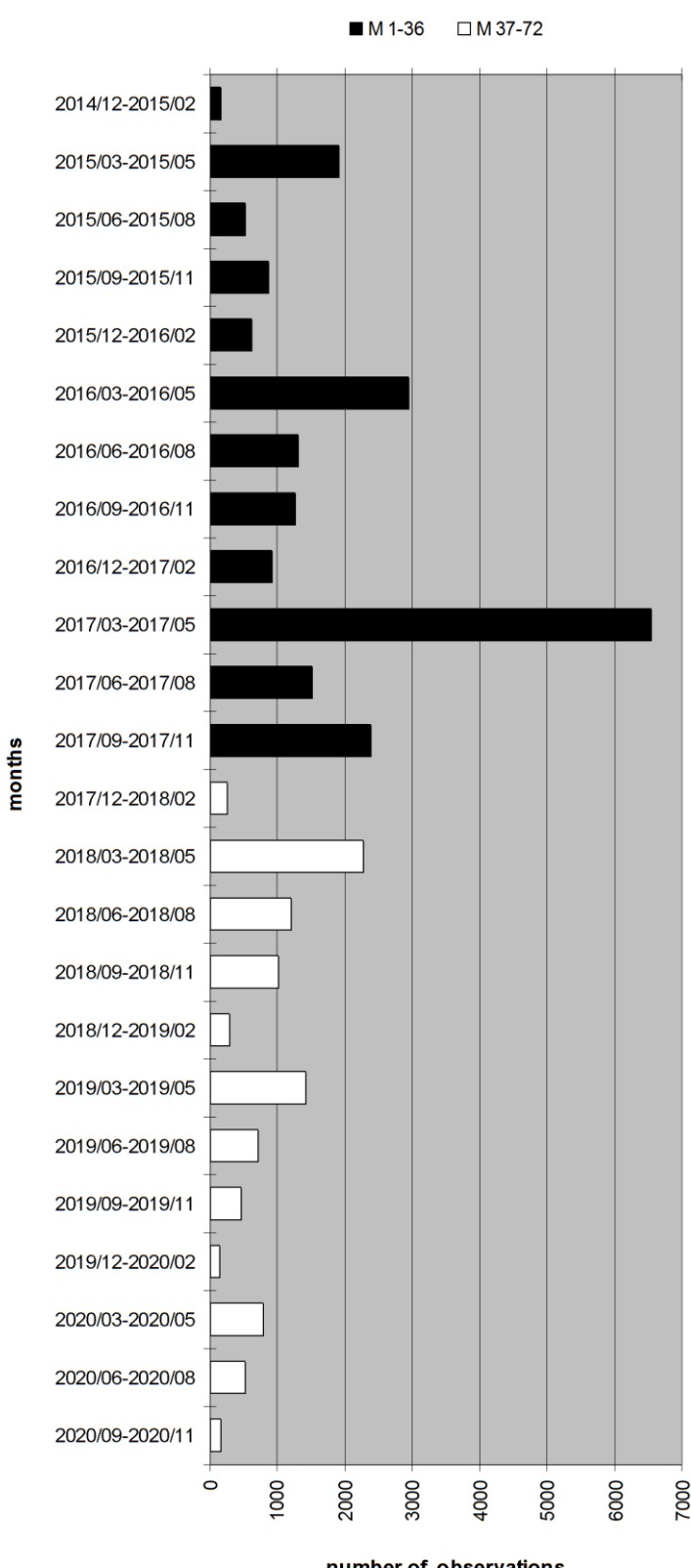

**Figure 1.** Trend of observations per three-month periods. M 1–36 (black bars) refer to the funded period, while M 37–72 (white bars) refer to the follow-up.

The distribution of volunteers in campaigns is reported in Figure 2. The most participated overall was "Chiedilo all'esperto" (Ask the expert), with ca. 560 volunteers,

followed by the "Licheni" (Lichens) and "Piante" (Plants), with ca. 300 and 200 volunteers, respectively. The most participated campaigns in M 37–72 were "Chiedilo all'esperto" (Ask the expert) and "Urban Nature" (ca. 100 volunteers). The only campaigns with similar participation in both periods were "Biodiversità Brescia GERT" (Biodiversity in Brescia) and "Urban Nature".

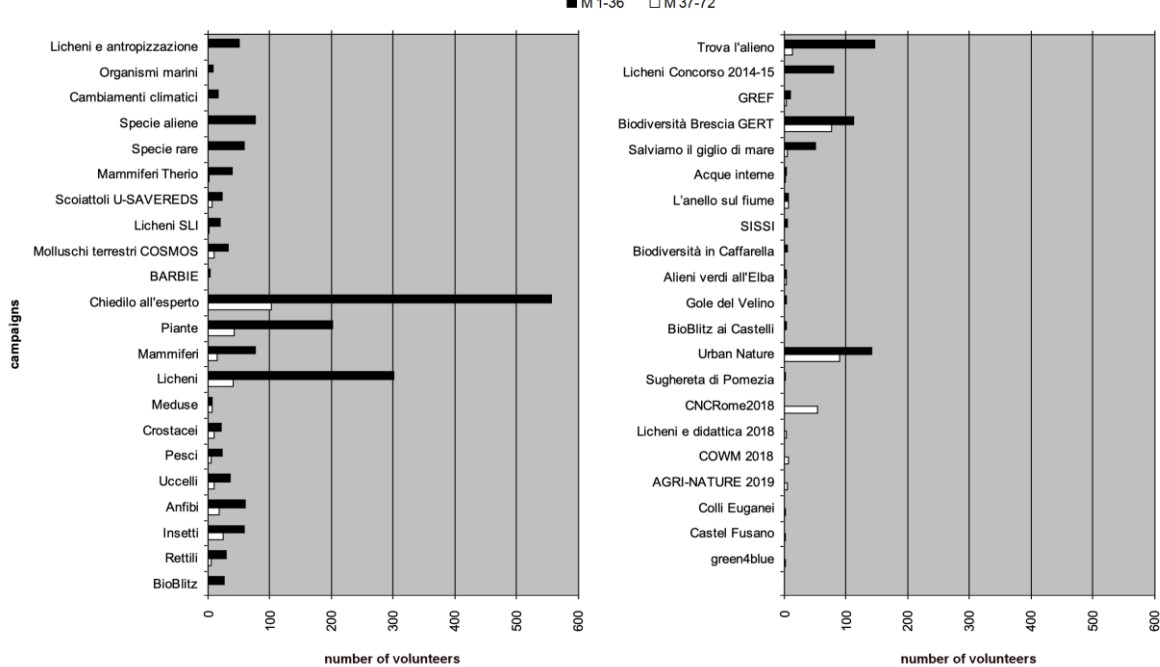

**Figure 2.** Number of volunteers for each campaign. M 1–36 (black bars) refer to the funded period, while M 37–72 (white bars) refer to the follow-up. The English names of the campaigns are provided in Table 2.

Table 2 reports the performance of each campaign in terms of the number of observations. Observations are reported for M 1–36, M 37–72, and overall. The percentage increase (if any) in M 37–72 is reported as well. Several campaigns were particularly active in M 37–72, with the campaign "Biodiversità Brescia GERT" (Biodiversity in Brescia) doubling the number of observations collected during M 1–36 (+203%). A few campaigns, and especially "CNCRome2018" (related to the "City Nature Challenge", 2018), were active in M 37–72 only. In Table 3, the months of activity (i.e., the number of months for which at least one observation was collected for the campaign) in M 1–36, M 37–72, and overall, and the range of activity (i.e., the number of months between the first and the last month during which at least one observation was collected) are reported, together with the starting and ending (or whether the campaign is ongoing). Only a limited amount of campaigns were active for their whole range, while several were active during limited periods. The only campaign which had at least one observation per month was "Biodiversità Brescia GERT" (Biodiversity in Brescia).

**Table 2.** Performances of campaigns.

| Campaign | | Number of Observations in Periods | | | Increase of Observations in M 37–72 (%) |
|---|---|---|---|---|---|
| ID | Name | M 1–36 | M 37–72 | Overall | |
| 1 | Licheni e antropizzazione (Lichens and anthropization) | 427 | | 427 | |
| 6 | Organismi marini (Marine organisms) | 24 | | 24 | |
| 13 | Cambiamenti climatici (Climate changes) | 28 | | 28 | |
| 14 | Specie aliene (Alien species) | 884 | | 884 | |
| 15 | Specie rare (Rare species) | 149 | | 149 | |
| 16 | Mammiferi Therio (Mammals Therio) | 101 | 2 | 103 | 2 |
| 17 | Scoiattoli U-SAVEREDS (Squirrels U-SAVEREDS) | 61 | 7 | 68 | 11 |
| 18 | Licheni SLI (Lichens SLI) | 34 | 1 | 35 | 3 |
| 21 | Molluschi terrestri COSMOS (Terrestrial molluscs) | 82 | 17 | 99 | 21 |
| 23 | BARBIE | 3 | | 3 | |
| 25 | Chiedilo all'esperto (Ask the expert) | 3836 | 738 | 4574 | 19 |
| 26 | Piante (Plants) | 1088 | 165 | 1253 | 15 |
| 27 | Mammiferi (Mammals) | 143 | 17 | 160 | 12 |
| 29 | Licheni (Lichens) | 6462 | 307 | 6769 | 5 |
| 30 | Meduse (Jellyfish) | 11 | 7 | 18 | 64 |
| 31 | Crostacei (Crustaceans) | 23 | 13 | 36 | 57 |
| 32 | Pesci (Fish) | 45 | 5 | 50 | 11 |
| 33 | Uccelli (Birds) | 75 | 17 | 92 | 23 |
| 34 | Anfibi (Amphibians) | 119 | 22 | 141 | 18 |
| 35 | Insetti (Insects) | 123 | 33 | 156 | 27 |
| 36 | Rettili (Reptiles) | 48 | 8 | 56 | 17 |
| 38 | BioBlitz | 468 | | 468 | |
| 39 | Trova l'alieno (Find the alien) | 1132 | 21 | 1153 | 2 |
| 40 | Licheni Concorso 2014-15 (Lichens Contest 2014-15) | 535 | | 535 | |
| 42 | GREF | 395 | 32 | 427 | 8 |
| 43 | Biodiversità Brescia GERT (Biodiversity in Brescia) | 2750 | 5572 | 8322 | 203 |
| 44 | Salviamo il giglio di mare (Let's save the sea daffodil) | 130 | 6 | 136 | 5 |
| 45 | Acque interne (Inland waters) | 7 | 1 | 8 | 14 |
| 46 | L'anello sul fiume (The ring on the river) | 120 | 10 | 130 | 8 |
| 50 | SISSI | 12 | | 12 | |
| 52 | Biodiversità in Caffarella (Biodiversity in Caffarella) | 86 | | 86 | |
| 53 | Alieni verdi all'Elba (Green aliens on Elba) | 5 | 5 | 10 | 100 |
| 54 | Gole del Velino (Gorges of Velino) | 54 | | 54 | |
| 56 | BioBlitz ai Castelli (BioBlitz at the Castelli) | 7 | | 7 | |
| 58 | Urban Nature | 1419 | 608 | 2027 | 43 |
| 60 | Sughereta di Pomezia | 2 | | 2 | |
| 61 | CNCRome2018 | | 1471 | 1471 | |
| 62 | Licheni e didattica 2018 (Lichens and teaching 2018) | | 15 | 15 | |
| 63 | COWM 2018 | | 48 | 48 | |
| 64 | AGRI-NATURE 2019 | | 13 | 13 | |
| 65 | Colli Euganei | | 7 | 7 | |
| 66 | Castel Fusano | | 4 | 4 | |
| 68 | green4blue | | 2 | 2 | |

M 1–36: project activity period; M 37–72: follow-up period. The English names of campaigns are also reported, when necessary.

Retention is reported in Table 4. Most volunteers contributed only for a limited amount of time—for 1 month or from 2 to 5 months (72.47% and 15.15%). Volunteers who sent a single observation were 402 (29.28%). Only 12.38% of volunteers were retained for more than 5 months, and 39 (2.84%) for more than 30 months. In general, the youngest volunteers (2001–2008) were mostly active for up to 5 months (97.83%). In all other age classes, the volunteers active for more than 5 months never dropped below 7%, rising to more than

20% in classes <=1950, 1951–1960, and 1961–1970. However, there are significant differences among age classes in each retention group (Table 4), mostly due to the "younger" and the "older" age classes. Except in the case of group 2–5, the retention in class 2001–2008 is significantly different from almost all other age classes (Appendix A, Table A1). This is especially true for retention group 1, where the class 2001–2008 is significantly different from all other classes, evidencing a more relevant "touch and go" behaviour in pupils. On the contrary, the oldest age class (<=1950) is significantly different from all other age classes in the higher retention group (>30 months of participation), thus possibly highlighting a higher commitment. Age class 1971–1980 is significantly different from the older classes in the lowest retention group (1 month), thus possibly evidencing a sort of "turning point" as far as age is concerned for the "touch and go" behaviour.

**Table 3.** Months of activity for each campaign.

| Campaign | | Months of Activity | | | Range of Activity | | |
|---|---|---|---|---|---|---|---|
| ID | Name | M 1–36 | M 37–72 | Overall | Months | Started | Ended |
| 1 | Licheni e antropizzazione | 13 | 0 | 13 | 17 | M1 | M20 |
| 6 | Organismi marini | 4 | 0 | 4 | 6 | M1 | M20 |
| 13 | Cambiamenti climatici | 5 | 0 | 5 | 7 | M1 | M20 |
| 14 | Specie aliene | 13 | 0 | 13 | 17 | M1 | M20 |
| 15 | Specie rare | 9 | 0 | 9 | 20 | M1 | M20 |
| 16 | Mammiferi Therio | 21 | 2 | 23 | 54 | M2 | ongoing |
| 17 | Scoiattoli U-SAVEREDS | 20 | 5 | 25 | 65 | M3 | ongoing |
| 18 | Licheni SLI | 7 | 1 | 8 | 36 | M4 | ongoing |
| 21 | Molluschi terrestri COSMOS | 25 | 11 | 36 | 61 | M6 | ongoing |
| 23 | BARBIE | 2 | 0 | 2 | 2 | M9 | ongoing |
| 25 | Chiedilo all'esperto | 28 | 29 | 57 | 63 | M10 | ongoing |
| 26 | Piante | 26 | 23 | 49 | 61 | M11 | ongoing |
| 27 | Mammiferi | 26 | 12 | 38 | 61 | M11 | ongoing |
| 29 | Licheni | 24 | 26 | 50 | 61 | M11 | ongoing |
| 30 | Meduse | 7 | 6 | 13 | 32 | M11 | ongoing |
| 31 | Crostacei | 15 | 9 | 24 | 63 | M11 | ongoing |
| 32 | Pesci | 13 | 5 | 18 | 63 | M11 | ongoing |
| 33 | Uccelli | 20 | 9 | 29 | 59 | M11 | ongoing |
| 34 | Anfibi | 25 | 10 | 35 | 56 | M11 | ongoing |
| 35 | Insetti | 23 | 14 | 37 | 61 | M11 | ongoing |
| 36 | Rettili | 14 | 5 | 19 | 47 | M11 | ongoing |
| 38 | BioBlitz | 2 | 0 | 2 | 24 | M11 | ongoing |
| 39 | Trova l'alieno | 25 | 10 | 35 | 56 | M11 | ongoing |
| 40 | Licheni Concorso 2014-15 | 9 | 0 | 9 | 11 | M11 | M21 |
| 42 | GREF | 16 | 7 | 23 | 47 | M14 | ongoing |
| 43 | Biodiversità Brescia GERT | 20 | 36 | 56 | 56 | M16 | ongoing |
| 44 | Salviamo il giglio di mare | 11 | 5 | 16 | 49 | M19 | ongoing |
| 45 | Acque interne | 3 | 1 | 4 | 26 | M20 | ongoing |
| 46 | L'anello sul fiume | 3 | 5 | 8 | 32 | M26 | ongoing |
| 50 | SISSI | 7 | 0 | 7 | 9 | M28 | ongoing |
| 52 | Biodiversità in Caffarella | 4 | 0 | 4 | 6 | M29 | ongoing |
| 53 | Alieni verdi all'Elba | 3 | 2 | 5 | 16 | M30 | ongoing |
| 54 | Gole del Velino | 3 | 0 | 3 | 3 | M30 | ongoing |
| 56 | BioBlitz ai Castelli | 2 | 0 | 2 | 2 | M30 | ongoing |
| 58 | Urban Nature | 2 | 16 | 18 | 38 | M30 | ongoing |
| 60 | Sughereta di Pomezia | 1 | 0 | 1 | 1 | M30 | M35 |
| 61 | CNCRome2018 | 0 | 9 | 9 | 14 | M40 | ongoing |
| 62 | Licheni e didattica 2018 | 0 | 4 | 4 | 7 | M46 | M56 |
| 63 | COWM 2018 | 0 | 1 | 1 | 1 | M48 | M49 |
| 64 | AGRI-NATURE 2019 | 0 | 3 | 3 | 3 | M54 | ongoing |
| 65 | Colli Euganei | 0 | 4 | 4 | 8 | M60 | ongoing |
| 66 | Castel Fusano | 0 | 2 | 2 | 2 | M60 | ongoing |
| 68 | green4blue | 0 | 1 | 1 | 1 | M71 | ongoing |

M 1–36: project activity period; M 37–72: follow-up period. The English names of the campaigns are provided in Table 2. For each campaign, other than the number of months of activity (i.e., those in which at least one observation has been reported), the months between the first and the last observation are reported as well. Furthermore, the beginning and ending (if not ongoing) months are reported.

**Table 4.** Volunteers divided into retention groups and age classes.

| Retention Groups | <=1950 | 1951–1960 | 1961–1970 | 1971–1980 | 1981–1990 | 1991–2000 | 2001–2008 | Total | $p$ |
|---|---|---|---|---|---|---|---|---|---|
| 1 | 16 57.14% | 70 560.87% | 128 562.75% | 201 575.28% | 111 566.87% | 201 574.17% | 268 583.23% | 995 572.47% | <0.001 |
| 2–5 | 1 53.57% | 17 514.78% | 32 515.69% | 28 510.49% | 33 519.88% | 50 518.45% | 47 514.60% | 208 515.15% | 0.047 |
| 6–10 | 4 514.29% | 7 56.09% | 10 54.90% | 9 53.37% | 8 54.82% | 7 52.58% | 2 50.62% | 47 53.42% | <0.001 |
| 11–20 | 2 57.14% | 11 59.57% | 16 57.84% | 18 56.74% | 8 54.82% | 3 51.11% | 4 51.24% | 62 54.52% | <0.001 |
| 21–30 | 0 50% | 3 52.61% | 8 53.92% | 6 52.25% | 2 51.20% | 3 51.11% | 0 50% | 22 51.60% | 0.023 |
| >30 | 5 517.86% | 7 56.09% | 10 54.90% | 5 51.87% | 4 52.41% | 7 52.58% | 1 50.31% | 39 52.84% | <0.001 |
| Total | 28 52.04% | 115 58.38% | 204 514.86% | 267 519.45% | 166 512.09% | 271 519.74% | 322 523.45% | 1373 5100% | |

Retention groups are based on the number of months in which each volunteer contributed to the project. Retention groups are 1 month, 2–5 months, 6–10 months, 11–20 months, 21–30 months, and >30 months. Age classes are based on the year of birth. The number of volunteers and the percentages for each age class are reported. The "Total" row and column show the percentages calculated for 1373 volunteers. The significance of the diversity of age classes per retention group was tested by mean of a Pearson's Chi-squared test.

Age class and retention seem to influence the correctness of the observations, with older volunteers achieving better performances (Table 5). In general, the percentage of correct observations is significantly higher in M 37–72 (Wilcoxon matched pairs test, $p < 0.05$). The increase in the number of observations in M 37–72 is higher for the oldest classes and lower for the youngest, for which retention from M 1–36 is the lowest (Table 5). Volunteers of different age classes have a significantly different performance in the two periods (Chi-squared test $p < 0.001$). Plus, the differences between age classes are almost all significant (Appendix A, Table A2). In particular, the two "older" age classes (<= 1950, 1951–1960) have a significant different performance in M 37–72. On the contrary, "younger" classes (1991–2000, 2001–2008) have a significant different performance in the M 1–36. The increase in volunteers per age classes in M 37–72 does not evidence a particular trend, but for a major increase in the oldest age class (Table 5). On the other hand, the longer the retention of volunteers, the higher the percentage of correct observations (Table 6).

**Table 5.** Effects of age classes.

| Age Classes | % of Correct Observations | | | Increase of Observations in M 37–72 (%) | Volunteers Retained from M 1–36 (%) | Increase of Volunteers in M 37–72 (%) |
|---|---|---|---|---|---|---|
| | M 1–36 | M 37–72 | Overall | | | |
| <=1950 | 86.63 | 97.99 | 96.74 | 724.42 | 31.25 | 62.50 |
| 1951–1960 | 89.58 | 98.96 | 94.82 | 121.07 | 6.59 | 24.44 |
| 1961–1970 | 88.97 | 85.09 | 88.04 | 36.66 | 11.18 | 31.58 |
| 1971–1980 | 74.56 | 82.30 | 76.56 | 34.34 | 4.57 | 35.53 |
| 1981–1990 | 85.77 | 91.15 | 86.95 | 27.63 | 10.00 | 26.15 |
| 1991–2000 | 78.77 | 86.30 | 81.22 | 46.89 | 4.05 | 18.47 |
| 2001–2008 | 63.56 | 77.75 | 65.69 | 17.69 | 2.51 | 33.05 |
| $p$ | <0.001 | <0.001 | | | | |

Age classes based on the year of birth. M 1–36: project activity period; M 37–72: follow-up period. The significance of the diversity in the ratio correct/wrong observations per project period in the different age classes was tested by mean of a Pearson's Chi-squared test.

The recruitment of new volunteers was not constant in time (Figure 3), with several bursts which could be related to effective promotional events. Among them there were: the broadcasting of CSMON-LIFE in two national television shows (April and October 2015), an article on a widespread national magazine (August 2015), two large school contests (April/May 2016 and 2017), an event in a public park in the centre of Rome (May 2017) which had major press coverage, two events ("Urban Nature", October 2017 and 2018) organized together with the Italian WWF, and the "City Nature Challenge" in April 2018.

**Table 6.** Effect of retention on the correctness of observations.

| Retention Groups | % of Correct Observations |
|:---:|:---:|
| 1 | 70.53 |
| 2–5 | 74.97 |
| 6–10 | 87.95 |
| 11–20 | 90.14 |
| 21–30 | 92.20 |
| >30 | 96.70 |

Retention groups are based on the number of months in which each volunteer contributed to the project. Retention groups are 1 month, 2–5 months, 6–10 months, 11–20 months, 21–30 months, and >30 months.

The "diversity" of observations is reported in Table 7. Even if these data should be taken carefully, since the possibility of identifying an organism at least at the genus level is different for different taxonomic groups, ca. 1800 genera were reported (with a rate of ca. of 1 genus every 17 observations). The most reported are the lichen genera *Xanthoria* and *Flavoparmelia* (*X. parietina* and *F. caperata* were target organisms in schools contests) and the genus *Ailanthus* (with the invasive alien plant *Ailanthus altissima*). Among the genera with at least 100 observations, those with the highest error rate were four lichens, *Parmotrema* (91%), *Diploicia* (86%), *Flavoparmelia* (55%), and *Evernia* (52%). Those with the lowest error rate were the plant genera *Pancratium* (0%), *Ruscus* and *Acer* (2%), and the invasive alien *Phytolacca* (3%), together with the bird genus *Anas* (1%).

**Table 7.** Diversity of the observations collected by volunteers.

| Genus | Observations | | | | | | |
|:---:|:---:|:---:|:---:|:---:|:---:|:---:|:---:|
| | Correct | | Wrong | | Uncertain | | Total |
| | n | % | n | % | n | % | n |
| *Xanthoria* | 3443 | 86% | 554 | 14% | 2 | 0% | 3999 |
| *Flavoparmelia* | 1519 | 45% | 1864 | 55% | 3 | 0% | 3386 |
| *Ailanthus* | 1428 | 93% | 115 | 7% | 0 | 0% | 1543 |
| *Robinia* | 862 | 91% | 86 | 9% | 0 | 0% | 948 |
| *Hedera* | 371 | 91% | 37 | 9% | 0 | 0% | 408 |
| *Evernia* | 174 | 47% | 193 | 52% | 3 | 1% | 370 |
| *Opuntia* | 340 | 94% | 21 | 6% | 0 | 0% | 361 |
| *Phytolacca* | 234 | 98% | 6 | 3% | 0 | 0% | 240 |
| *Trachemys* | 211 | 92% | 15 | 7% | 4 | 2% | 230 |
| *Parmotrema* | 16 | 7% | 199 | 91% | 4 | 2% | 219 |
| *Podarcis* | 166 | 92% | 14 | 8% | 1 | 1% | 181 |
| *Psittacula* | 133 | 80% | 31 | 19% | 3 | 2% | 167 |
| *Ruscus* | 161 | 98% | 4 | 2% | 0 | 0% | 165 |
| *Crataegus* | 94 | 60% | 63 | 40% | 0 | 0% | 157 |
| *Diploicia* | 22 | 14% | 135 | 86% | 0 | 0% | 157 |
| *Quercus* | 143 | 92% | 13 | 8% | 0 | 0% | 156 |
| *Myopsitta* | 139 | 90% | 10 | 6% | 6 | 4% | 155 |
| *Pancratium* | 155 | 100% | 0 | 0% | 0 | 0% | 155 |
| *Vanessa* | 103 | 71% | 40 | 28% | 2 | 1% | 145 |
| *Apis* | 84 | 58% | 59 | 41% | 1 | 1% | 144 |
| *Carpobrotus* | 109 | 89% | 12 | 10% | 1 | 1% | 122 |
| *Acer* | 117 | 98% | 2 | 2% | 0 | 0% | 119 |
| *Pinus* | 83 | 74% | 29 | 26% | 0 | 0% | 112 |
| *Anas* | 102 | 99% | 1 | 1% | 0 | 0% | 103 |
| *Sciurus* | 94 | 91% | 8 | 8% | 1 | 1% | 103 |

Only genera with at least 100 observations are reported.

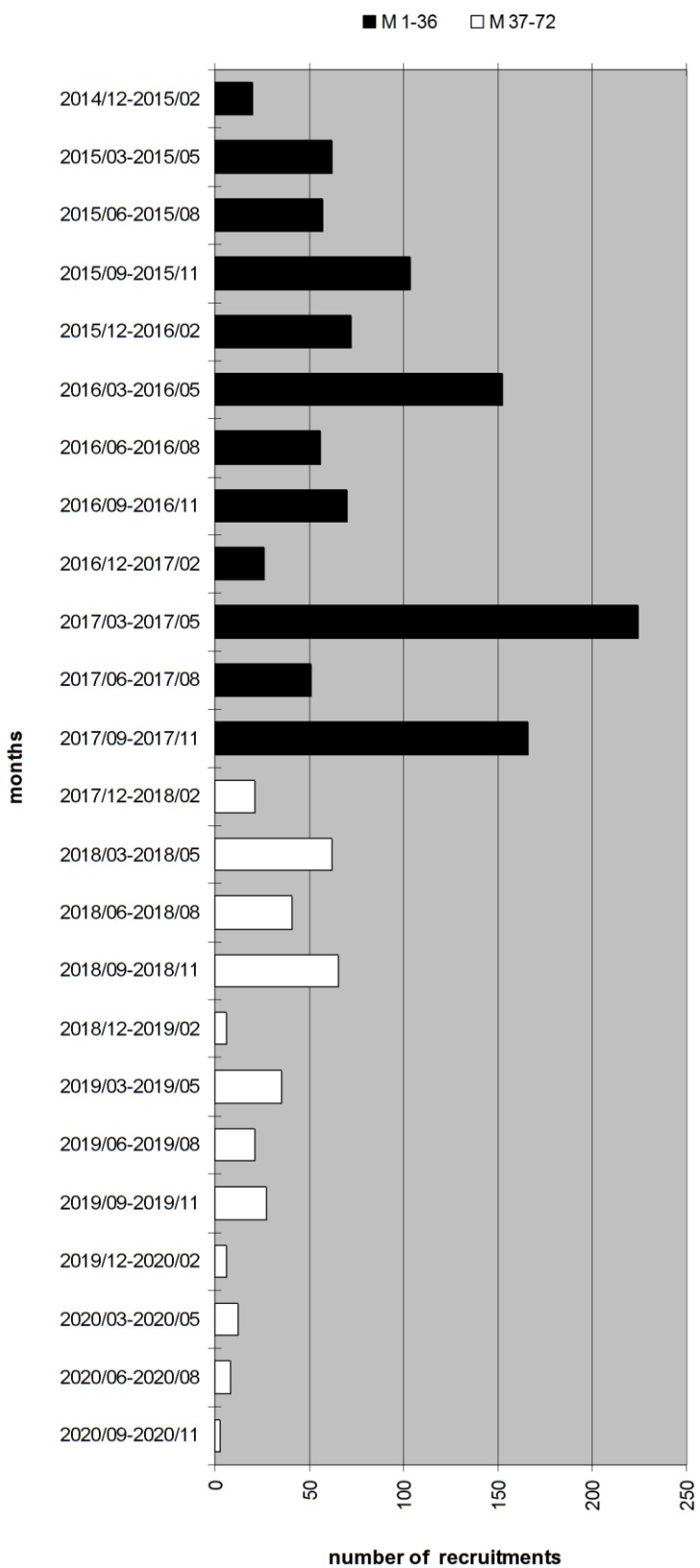

**Figure 3.** Trend of new recruitments per three-month periods. M 1–36 (black bars) refer to the funded period, while M 37–72 (white bars) refer to the follow-up.

## 4. Discussion

CSMON-LIFE was funded in the framework of the LIFE+ 2007–2013 programme. As in the case of several other projects, communication and awareness-raising initiatives, given their cost, are normally possible during the funded period only. Even if projects should maintain outputs (such as web pages, apps, etc.) active and available even after the funded period, the actual impact of the two periods (funded and follow-up) could be extremely different. Thus, we aimed at understanding the impact of CSMON-LIFE during the funded (M 1–36) and follow-up period ("after-LIFE", M 37–72). In CSMON-LIFE, the recruitment of volunteers was dependent on the effectiveness of the awareness-raising and communication actions, for which no funding was available during M 37–72. Thus, both the retention of volunteers and further recruitment were based on the satisfaction of participants and word of mouth. M 37–72 were less participated (Table 1). Since the funded period, 5.25% of the volunteers were retained, while 367 new volunteers were recruited mostly thanks to the dissemination by those who were retained and to the adoption of CSMON-LIFE as the official platform for the first Italian participation in the "City Nature Challenge" (https://citynaturechallenge.org, accessed on 26 April 2021) and in the second edition of the WWF initiative "Urban Nature" (the first one was held during M 1–36). While the number of volunteers during M 37–72 was 29.97% of M 1–36, the number of observations was 43.92%, thus highlighting a possibly higher level of commitment. This could be supported by the increased rate of correctness for the observations (Table 1). At least for the funded period, these data can be compared to those of another LIFE+ project, LIFE MIPP (Monitoring Insects with Public Participation), which operated in Italy during the same period [36]. The rate of correctness reported from MIPP (73%) is close to M 1–36 (78%). However, to our knowledge, no follow-up data are available either for MIPP or other projects.

The performance of volunteers, measured as the percentage of correct observations, seems to be influenced by age. During M 37–72, volunteers were recruited especially in the "older" age classes, with the lowest increase in the 1991–2000 age class (Table 5), while the increase in the 2000–2008 age class could be an effect of the school contest "Licheni e Didattica 2018" (Lichens and Teaching 2018). In general, the involvement of schools leads to the recruitment of large amounts of volunteers, which are generally retained, however, for a limited time (Table 4), and often report one observation only before quitting. Campaigns involving especially—if not only—schools, i.e., "Licheni" (Lichens) and "Trova l'alieno" (Find the alien) (Tables 2 and 3), were especially active during M 1–36, with a limited increase in observation (5% and 2%) during M 37–72.

In this study, retention is measured as the number of months in which a volunteer was active (i.e., sent observations). If we consider "not retained" volunteers who sent a single observation only, the retention rate in CSMON-LIFE is ca. 70%, far higher than other similar initiatives. On the contrary, the retention rate is lower (5.25%) if volunteers are considered retained when their activity spans from M 1–36 to M 37–72. This datum is comparable to other Citizen Science initiatives, even if comparisons are difficult because of the different time frames and an overall lack of follow-up data. In CSMON-LIFE, ca. 72% of volunteers were active for 1 month only (Table 4), and ca. 12% were active for more than 5 months. However, differences arise taking into account age classes. Volunteers active for 1 month only are significantly more in the youngest age class (Table 4 and Appendix A, Table A1), for which retention is significantly different than in all other age classes (Appendix A, Table A1). On the contrary, higher retention rates are evident in older age classes. Even if older volunteers are often more difficult to recruit, once engaged, they normally continue their activity for longer periods. This could suggest a higher commitment, possibly due to a more conscious participation, while at younger ages participation could be seen as a temporary "game" or as a sort of forced commitment when part of school activities, thus making a continued participation less probable. Brouwer and Hessels [26] reported that younger participants (aged <24 years) were relatively less eager to participate again (79%). Similarly, in CSMON-LIFE, both a limited retention and a poor increase of observations

were evidenced for the class 2000–2008 in M 37–72 (Table 5). Parrish et al. [37] observed that online Citizen Science projects usually have a low retention, and the turnover of participants can sometimes be close to 100% per year. Plus, single local events as "BioBlitzes", even if successful, tend to attract more dabblers, leading to a low retention rate. On the other end, high retention projects are those which attract less dabblers and target individuals which have a relevant interest in the mission of a project. Jacobson et al. [38] highlighted that projects with high retention rates usually have a good leadership, clear expectations, and meaningful tasks. As an example, a high retention rate (ca. 38%) is reported by Ang et al. [33] for a project including repeated field work, in which training and support were provided to volunteers throughout the project. CSMON-LIFE was mostly based on online activities, as well as spot events, such as "BioBlitzes", "Scoprinatura" (Discover nature), etc., lasting 1 to few days, with no or little training provided to the volunteers, and thus a low retention rate was expected. However, the retention for more than 10 months was relatively high, close to 9%.

Age class seems to influence the correctness of observations (Table 5). The youngest class, 2000–2008, has the lowest correctness rate, while the two older classes have the highest. The differences are almost always significant (Appendix A, Table A2), with a trend of increase in the rate of correct observations with the age of participants, but for the class 1971–1980. The error rate of the observations collected by students is higher, as visible in the high error rate for the taxa which were used as targets in the contests for schools (*Diploicia*, 86%, *Flavoparmelia*, 55%, *Evernia*, 52%, and *Parmotrema*, 91%, Table 7). However, the rate of correct observations increases for almost all age classes in M 37-72 (Table 5). This is probably due to a higher commitment by the volunteers recruited in the follow-up, or retained since M 1–36. The rate of correct observations also increases with retention (Table 6). Volunteers active for 1 month only are those with the highest error rate, while those who continued their activity for more than 10 months have an error rate of less than 10%. Thus, it is safe to assume that an increased experience leads to an overall better performance. Not all the campaigns of CSMON-LIFE were successful (Tables 2 and 3). Since the beginning, several "main" campaigns were issues, followed by several "guest" campaigns. Guest campaigns were developed upon request by citizens or other initiatives/projects, such as "Biodiversità Brescia GERT" (Biodiversity in Brescia) and "Urban Nature", or for events such as BioBlitzes, contests, etc. Several guest campaigns were issued during M 37–72, such as "CNCRome2018". While all active campaigns collected observations even during M 37–72, their performance ranges from very slight increases to more than doubling the observations collected in M 1–36 (Table 2). The main campaigns had a general increase in observations from 11% to 27%, and the one which collected more observations was "Chiedilo all'esperto" (Ask the expert). The interest in this campaign is probably due to the curiosity of many citizens about biodiversity in anthropized areas, with most observations reporting insects and plants which are common in the urban environment. Among the guest campaigns, the most successful are "Biodiversità Brescia GERT" (Biodiversity in Brescia), with a total of 8322 observations, of which 5572 in M 37–72, "Urban Nature", with 2027 observations, of which 608 in M 37–72, and "CNCRome2018", with 1471 observations (Table 2). "Biodiversità Brescia GERT" (Biodiversity in Brescia) is also the only campaign which collected at least one observation for each month. The others collected observations only during a portion of their range (Table 3). The discrepancy between the months of activity and the duration of the campaigns can be related to the seasonality of many volunteers. In fact, the flow of observations was not constant, but had two relevant bursts each year (spring and autumn). While the first burst can be related to the favourable season, in which the phenology of most organisms makes them more visible, the autumn burst could be related mostly to specific project activities (e.g., the beginning of school contests). A similar trend could be evidenced in the recruitment of volunteers (Figure 3), which can be related to specific events, such as "Urban Nature" and "City Nature Challenge", to school contests, and, at least during the M 1–36 period, to particularly effective communication actions.

As far as recruitment is concerned, there is a slight trend of increase in the number of volunteers from the older to the younger class (Table 4). However, the high numbers of the class 2000–2008 are mostly due to initiatives such as school contests. Brouwer and Hessels [26] evidenced that often, in the absence of specific targeted engagement strategies, the most represented age classes are the younger ones, possibly since they have a more continuous access to social media. On the contrary, target invitation strategies lead to the recruitment of older age classes. CSMON-LIFE communication and awareness activities focused on different media, from social (Facebook especially) to "classic" (television, newspapers, magazines), and this could explain the recruitment of volunteers from different age classes.

The performance of one of the most successful campaigns of the project, "Biodiversità Brescia GERT" (Biodiversity in Brescia), requires some further considerations. This guest campaign (started in April, 2016) was developed upon request by a group of citizens in the municipality of Brescia (in Northern Italy), with the aim of demonstrating to local decision-makers the relevance of several abandoned gravel pits for biodiversity conservation, in order to avoid turning them into waste landfills. This is the only case of an actual bottom-up, citizen-driven action in the project. Its relevant success, with the collection of 9294 observations as of September 29, 2021, highlights the relevance of a strong commitment from volunteers.

On the basis of performance (in terms of error rate) and retention, it seems that older age participants, once engaged, are more motivated than younger ones. Contrasting results are reported by other studies. Larson et al. [24] reported that the volunteers were mostly older than the American average, highly educated, and with averagely high incomes. On the other hand, Strasser et al. [4] reported that surveys from several participatory projects indicate that the volunteers are primarily white, younger than average, middle class, and men. Asingizwe et al. [23] reported an equal number of males and female participants, and a similar number in the age classes <35 and >35 years old. In general, West and Pateman [39] highlight that relatively little research is available on what influences volunteers' participation or what encourages them to continue their engagement. From the CSMON-LIFE experience, it seems that the involvement of schools can lead to large numbers in terms of engagement, but also to little retention and poorer data quality (higher error rate) than the involvement of more mature citizens.

## 5. Conclusions

CSMON-LIFE was a Governance and Information LIFE+ project aimed at improving the effectiveness of governance by increasing the number of citizens who provide reliable environmental data. Communication and awareness-raising actions aimed at recruiting volunteers were not targeted to a specific target group, nor did they adopt a single medium, thus being generalist in nature. The volunteers' distribution in different age classes seems to highlight that—at least in Italy—environmental issues are of wide interest among citizens. Plus, the use of modern technologies, such as smartphones and apps, does not seem to constitute a barrier for the participation of older citizens. The performance of different campaigns, and especially the success of "Biodiversità Brescia GERT" (Biodiversity in Brescia), seems to highlight that volunteers are more motivated when involved since the development of a campaign (or when they are its actual driver, in an actual bottom-up strategy), and when dealing with issues which are not "global" (biodiversity, global change, etc.) but "local" ("our" biodiversity, the effects of global change in "our" municipality, etc.). In particular, the a priori involvement of volunteers, since the development of a Citizen Science campaign, seems to generate a stronger motivation. This should be taken into consideration when developing future Citizen Science projects.

CSMON-LIFE, together with other Citizen-Science-oriented actions in Italy (such as other LIFE project, e.g., MIPP, http://lifemipp.eu/mipp/new/ and U-SAVEREDS, http://usavereds.eu/it_IT/, accessed on 26 April 2021), led to an increased interest on Citizen Science among several stakeholders, which are now willing to include Citizen Science

approaches in their activities. The inclusion of Citizen Science Data in the National Biodiversity Network, and the adoption of digital tools for supporting citizens in the collection of environmental data by several regional observatories of biodiversity, are probably the first steps towards promoting Citizen Science in the country. This is particularly interesting as regards the potential of Citizen Science and Citizen Science Data for complementing the official statistics used for reporting the degree of achievement of several UN Sustainable Development Goals.

**Author Contributions:** Conceptualization, S.M., F.A. and E.P.; methodology, S.M., D.C., A.M., D.P., M.P., G.T., F.A. and E.P.; validation, S.M., V.S. and E.P.; formal analysis, S.M. and E.P.; writing—original draft preparation, S.M.; writing—review and editing, D.C., A.M., D.P., M.P., G.T., V.S., E.P. and F.A.; visualization, E.P.; supervision, S.M.; project administration, S.M.; funding acquisition, S.M. All authors have read and agreed to the published version of the manuscript.

**Funding:** This research was funded by the European Commission in the framework of the LIFE+ programme (LIFE13 ENV/IT/842).

**Institutional Review Board Statement:** Not applicable.

**Informed Consent Statement:** Not applicable.

**Data Availability Statement:** Not applicable.

**Acknowledgments:** The authors are grateful to Oliviero Spinelli and the company Comunità Ambiente for their support during CSMON-LIFE and their suggestions on the manuscript.

**Conflicts of Interest:** The authors declare no conflict of interest. The funders had no role in the design of the study; in the collection, analyses, or interpretation of data; in the writing of the manuscript, or in the decision to publish the results.

## Appendix A

**Table A1.** Comparison between age classes for each retention group.

| R > 30 | <=1950 | 1951–1960 | 1961–1970 | 1971–1980 | 1981–1990 | 1991–2000 |
|---|---|---|---|---|---|---|
| 1951–1960 | **0.04396** | | | | | |
| 1961–1970 | **0.008949** | 0.651 | | | | |
| 1971–1980 | **<0.0001** | **0.03031** | 0.06356 | | | |
| 1981–1990 | **0.0003245** | 0.1181 | 0.2114 | 0.7033 | | |
| 1991–2000 | **<0.0001** | 0.09218 | 0.1781 | 0.5769 | 0.9106 | |
| 2001–2008 | **<0.0001** | **<0.0001** | **0.0003361** | 0.06018 | **0.02914** | **0.01687** |

| R = 21–30 | <=1950 | 1951–1960 | 1961–1970 | 1971–1980 | 1981–1990 | 1991–2000 |
|---|---|---|---|---|---|---|
| 1951–1960 | 0.3877 | | | | | |
| 1961–1970 | 0.2862 | 0.5372 | | | | |
| 1971–1980 | 0.4229 | 0.8308 | 0.289 | | | |
| 1981–1990 | 0.5593 | 0.3814 | 0.109 | 0.4335 | | |
| 1991–2000 | 0.5758 | 0.2754 | **0.0435** | 0.3025 | 0.9257 | |
| 2001–2008 | NA | **0.003634** | **0.0003425** | **0.006856** | **0.04842** | 0.05838 |

| R = 11–20 | <=1950 | 1951–1960 | 1961–1970 | 1971–1980 | 1981–1990 | 1991–2000 |
|---|---|---|---|---|---|---|
| 1951–1960 | 0.6893 | | | | | |
| 1961–1970 | 0.8967 | 0.5957 | | | | |
| 1971–1980 | 0.936 | 0.3392 | 0.6471 | | | |
| 1981–1990 | 0.607 | 0.1192 | 0.2402 | 0.413 | | |
| 1991–2000 | **0.01773** | **<0.0001** | **0.0002085** | **0.000741** | **0.0162** | |
| 2001–2008 | **0.02104** | **<0.0001** | **0.0001148** | **0.0004585** | **0.01564** | 0.8793 |

**Table A1.** *Cont.*

| R = 6–10 | <=1950 | 1951–1960 | 1961–1970 | 1971–1980 | 1981–1990 | 1991–2000 |
|---|---|---|---|---|---|---|
| 1951–1960 | 0.1443 | | | | | |
| 1961–1970 | 0.05054 | 0.651 | | | | |
| 1971–1980 | **0.007425** | 0.2241 | 0.4027 | | | |
| 1981–1990 | 0.05442 | 0.6421 | 0.9707 | 0.4505 | | |
| 1991–2000 | **0.001737** | 0.09218 | 0.1781 | 0.5907 | 0.2127 | |
| 2001–2008 | **<0.0001** | **0.0003961** | **0.001355** | **0.01413** | **0.001927** | 0.05158 |

| R = 2–5 | <=1950 | 1951–1960 | 1961–1970 | 1971–1980 | 1981–1990 | 1991–2000 |
|---|---|---|---|---|---|---|
| 1951–1960 | 0.1088 | | | | | |
| 1961–1970 | 0.08526 | 0.8298 | | | | |
| 1971–1980 | 0.2423 | 0.2322 | 0.09354 | | | |
| 1981–1990 | **0.03576** | 0.272 | 0.2918 | **0.006304** | | |
| 1991–2000 | **0.04629** | 0.3842 | 0.4301 | **0.008716** | 0.7116 | |
| 2001–2008 | 0.1038 | 0.9613 | 0.7331 | 0.1364 | 0.1353 | 0.2063 |

| R = 1 | <=1950 | 1951–1960 | 1961–1970 | 1971–1980 | 1981–1990 | 1991–2000 |
|---|---|---|---|---|---|---|
| 1951–1960 | 0.718 | | | | | |
| 1961–1970 | 0.5667 | 0.7403 | | | | |
| 1971–1980 | **0.03841** | **0.004431** | **0.003308** | | | |
| 1981–1990 | 0.3168 | 0.3018 | 0.4096 | 0.05783 | | |
| 1991–2000 | 0.05453 | **0.008973** | **0.007557** | 0.7668 | 0.1011 | |
| 2001–2008 | **<0.0001** | **<0.0001** | **<0.0001** | **0.01711** | **<0.0001** | **0.006881** |

Retention groups (R) based on the number of months in which each volunteer contributed to the project. Age classes are based on the year of birth. Significant *p*-values are highlighted in bold. Significance was tested by a mean of Pearson's Chi-squared test.

**Table A2.** Comparison of observations correctness rate between age classes in the two project periods.

| M 1–36 | <=1950 | 1951–1960 | 1961–1970 | 1971–1980 | 1981–1990 | 1991–2000 |
|---|---|---|---|---|---|---|
| 1951–1960 | 0.2235 | | | | | |
| 1961–1970 | 0.3526 | 0.5113 | | | | |
| 1971–1980 | **0.0003827** | **<0.0001** | **<0.0001** | | | |
| 1981–1990 | 0.7661 | **0.0009178** | **0.01023** | **<0.0001** | | |
| 1991–2000 | **0.01373** | **<0.0001** | **<0.0001** | **0.0003028** | **<0.0001** | |
| 2001–2008 | **<0.0001** | **<0.0001** | **<0.0001** | **<0.0001** | **<0.0001** | **<0.0001** |

| M 37–72 | <=1950 | 1951–1960 | 1961–1970 | 1971–1980 | 1981–1990 | 1991–2000 |
|---|---|---|---|---|---|---|
| 1951–1960 | **0.0107** | | | | | |
| 1961–1970 | **<0.0001** | **<0.0001** | | | | |
| 1971–1980 | **<0.0001** | **<0.0001** | 0.1406 | | | |
| 1981–1990 | **<0.0001** | **<0.0001** | **0.008951** | **0.0002353** | | |
| 1991–2000 | **<0.0001** | **<0.0001** | 0.4631 | **0.01168** | **0.02242** | |
| 2001–2008 | **<0.0001** | **<0.0001** | **0.001679** | 0.05015 | **<0.0001** | **<0.0001** |

M 1–36: project activity period; M 37–72: follow-up period. Age classes are based on the year of birth. Significant *p*-values are highlighted in bold. Significance was tested by mean of a Pearson's Chi-squared test.

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
