# Peer review of "Volunteers Recruitment, Retention, and Performance during the CSMON-LIFE (Citizen Science MONitoring) Project and 3 Years of Follow-Up"

_sustainability, doi:10.3390/su131911110_

Round 1
Reviewer 1 Report
Dear Authors,
despite the manuscript presented has a weak link with the theme of environmental sustainability, I believe that it provides an interesting contribution in the field of Citizen Science, and that it deserves to be published.
Follow-up data on volunteers' retention after the conclusion of CS projects are scarce in the literature, and they can be very useful to scholars to improve their work. I found the observations on seasonality and the motivations behind peaks in participation particularly interesting. The retention values linked to age groups are also valuable.
The structure of the paper is well constructed, the methods and results are generally presented clearly. The language and writing style are fluent.
However, I suggest making some small changes to increase the comprehensibility and usability of data. I list them below, along with some hints for improvement.
First of all, figures 1 and 3 are not readable due to poor resolution. Figure 2 is missing. In Figure 1 I recommend also improving the caption.
Although in the abstract you can read that the purpose of Life is "increasing awareness on CS in Italy", there is no clear geographical delimitation of the described activities. Did all CSMON-Life campaigns extend to the Italian territory? Were some campaigns only local? Was it possible to collect information outside the Italian borders? I suggest you explain the geographical extent of the activities in the introductory chapter (lines 92-97).
I also recommend adding web references to the project and campaigns mentioned, possibly where in the text they appear for the first time. In particular, references are never given to the CSMON-Life website and to the data portal (line 118).
The results shown are useful, because the retention of volunteers at the end of a project is rarely quantified. However, the usability of the data is limited by the fact that information about the "opening" period of each campaigns is not provided. In practice it is not possible to know if the volunteers stopped participating for personal choice or because the period in which they could send comments was over, or because the season was not favorable for observation. For example, in Table 3 it is not very useful to know the "range of activity" from the first to the last observation, while it would be useful to know the months of opening of the campaigns. Since it is not easy to express this complexity in a numerical table, I suggest using charts and time bars (some examples: https://www.datawrapper.de/tables), which can also be added as an appendix.
In the period M37-72 seven new campaigns were activated (see Table 2). Is it correct to consider them as a follow-up or are they rather new CS activities? Even if the same platform has been used, I think that the data resulting from these activities should be kept separate from the others so as not to create a bias in the retention rate.
In Table 4 and Table 6 please better explain the y axis "retention groups" (or at least specify their unit of measurement).
Line 437: “Brescia (Lombardy, N Italy)” is not a common wording. I suggest “Brescia, in northen Italy”; or “Brescia (in the Italian region of Lombardy)”; or just “Brescia“.
Author Response
We are grateful to the reviewer for the useful comments. We hope we have provided satisfying answers to all the issues.
Here below our answer to each comment.
Best regards,
Stefano Martellos
Q: First of all, figures 1 and 3 are not readable due to poor resolution. Figure 2 is missing. In Figure 1 I recommend also improving the caption.
A: Figures have been reviewed, and rebuild inverting the two axes. Thus, they are now easily readable. Figure 1 caption has been improved as well.
Q: Although in the abstract you can read that the purpose of Life is "increasing awareness on CS in Italy", there is no clear geographical delimitation of the described activities. Did all CSMON-Life campaigns extend to the Italian territory? Were some campaigns only local? Was it possible to collect information outside the Italian borders? I suggest you explain the geographical extent of the activities in the introductory chapter (lines 92-97).
A: as the reviewer noticed, even if campaigns were local, or limited to Italy, Citizen Scientist from abroad could report observations as well, simply downloading the app and registering. We decided not to impose any limitation, since our goal was that of maximizing the impact of awareness raising as far as the relevance of Citizen Science is concerned. We modified the introduction in order to highlight this.
Q: I also recommend adding web references to the project and campaigns mentioned, possibly where in the text they appear for the first time. In particular, references are never given to the CSMON-Life website and to the data portal (line 118).
A: We added the URL of the CSMON-LIFE web portal in the introduction.
Q: The results shown are useful, because the retention of volunteers at the end of a project is rarely quantified. However, the usability of the data is limited by the fact that information about the "opening" period of each campaigns is not provided. In practice it is not possible to know if the volunteers stopped participating for personal choice or because the period in which they could send comments was over, or because the season was not favorable for observation. For example, in Table 3 it is not very useful to know the "range of activity" from the first to the last observation, while it would be useful to know the months of opening of the campaigns. Since it is not easy to express this complexity in a numerical table, I suggest using charts and time bars (some examples: https://www.datawrapper.de/tables), which can also be added as an appendix.
A: given that reviewer 2 asked us to shorten the manuscript, we avoided adding further graphs. However, we improved table 3 adding two new columns, i.e. the month in which each campaign started, and the month in which it ended, or if it is ongoing. Plus, we improved its caption to make it more understandable. Some of the campaigns issued at the beginning of the project (M1) were converted into others in M11, and closed in M20, after the volunteers were informed and “migrated” in the new ones. But a few campaigns, devoted to specific events, all the others are still ongoing. Thus, we are safe to assume no volunteers ended their engagement in the project because of a campaign ending.
Q: In the period M37-72 seven new campaigns were activated (see Table 2). Is it correct to consider them as a follow-up or are they rather new CS activities? Even if the same platform has been used, I think that the data resulting from these activities should be kept separate from the others so as not to create a bias in the retention rate.
A: Actually, even if volunteers participated to novel campaigns only, this would not led to a bias in the retention rate, since it is measured for each volunteer, and counting the months of activity from the first observation to the last, whichever the campaign. Furthermore, among the novel campaigns in M37-72, one was the continuation of a previous one (Licheni e Didattica 2018), which continued the activity of the campaigns Licheni Concorso 2014-15 and Licheni. Thus, it could be hardly considered a novel one. The others were campaign of a very limited impact on the project, and led to the collection of a very limited number of observations, ca. 100 (see table 2).
Q: In Table 4 and Table 6 please better explain the y axis "retention groups" (or at least specify their unit of measurement).
A: We added a line to both captions in order to explain what the retention groups are.
Q: Line 437: “Brescia (Lombardy, N Italy)” is not a common wording. I suggest “Brescia, in northen Italy”; or “Brescia (in the Italian region of Lombardy)”; or just “Brescia“.
A: We modified the text as request
Reviewer 2 Report
The authors have done a commendable study on the subject of volunteer recruitment and retention. This is an important study that contributes to a little published area. Many organizations now include volunteers in varying degrees in their studies but relatively few bother to analyze the follow-up of their activities. Most that do report usually focus on the quality of the data collated by the volunteer programs. The combination with Citizen Science makes it that much more important.
The language, including spelling, in the paper, is poor and needs to be edited professionally.
Also, the paper is extremely wordy and can be shortened in the editing process by at least 20%.
Authors should concentrate all data in the Results section and discuss their implications in the Discussion. See lines 101-103 in the introduction as an example.
Figure 1 is simply too cluttered because of the time frame involved. Recommend the authors to coalesce the data into larger time blocks without losing the visual effect.
The same for Fig 2. Poor quality owing to the numerous campaigns included in the X-axis. Recommend to lump according to subjects or some other category such that the graph is readable. This figure is extraneous and can be deleted because all that data is given in Table 2.
Do not start a sentence with a number – line 207. Either rephrase such that the number is in the middle or spell out the number.
Table 4 is extraneous and I do not see the need for it. The data can be presented in the text.
Table 6 is also extraneous and the data can be presented in the text.
Figure 3 – same comment as Fig 1. Lump the dates without losing the visual impact.
Table 7 – add the footnote to the caption.
Line 290 – suggest changing after-LIFE to post-LIFE.
Author Response
We are grateful to the reviewer for the useful comments. We hope we have provided satisfying answers to all the issues.
Here below our answer to each comment.
Best regards,
Stefano Martellos
Q: The language, including spelling, in the paper, is poor and needs to be edited professionally. Also, the paper is extremely wordy and can be shortened in the editing process by at least 20%.
A: the had the manuscript checked by an English speaker, and reviewed it in order to make it less verbose. However, given the opinion of the other reviewer, we also tried to avoid shortening it too much. We hope our efforts could satisfy the reviewer’s request, while achieving a balance between the reviewers' opinions.
Q: Authors should concentrate all data in the Results section and discuss their implications in the Discussion. See lines 101-103 in the introduction as an example.
A: we have thoroughly rewritten this part of the introduction, moving the data in the results section.
Q: Figure 1 is simply too cluttered because of the time frame involved. Recommend the authors to coalesce the data into larger time blocks without losing the visual effect.
A: As in the answers for the first reviewer, we have changed figures 1, 2, and 3, orienting them vertically, and thus making them more readable. We hope this solution could satisfy the reviewer’s request.
Q: The same for Fig 2. Poor quality owing to the numerous campaigns included in the X-axis. Recommend to lump according to subjects or some other category such that the graph is readable. This figure is extraneous and can be deleted because all that data is given in Table 2.
A: As in the answers for the first reviewer, we have changed figures 1, 2, and 3, orienting them vertically, and thus making them more readable. We hope this solution could satisfy the reviewer’s request. Figure 2 and Table 2 report different information. The figure reports the number of volunteers per campaign, while the table reports the number of observations. Thus, we prefer to retain both in the manuscript.
Q: Do not start a sentence with a number – line 207. Either rephrase such that the number is in the middle or spell out the number.
A: we have addressed the issue.
Q: Table 4 is extraneous and I do not see the need for it. The data can be presented in the text. Table 6 is also extraneous and the data can be presented in the text.
A: While the data could be embedded in the text, we would prefer to retain the two table in the manuscript, since we this they are useful for highlighting the data. We hope the reviewer will allow this.
Figure 3 – same comment as Fig 1. Lump the dates without losing the visual impact.
A: As in the answers for the first reviewer, we have changed figures 1, 2, and 3, orienting them vertically, and thus making them more readable. We hope this solution could satisfy the reviewer’s request.
Q: Table 7 – add the footnote to the caption.
A: there is a footnote. Could it possibly went lost in the reviewer’s file?
Q: Line 290 – suggest changing after-LIFE to post-LIFE.
A: Given that the official definition for the follow-up periods in LIFE projects if after-LIFE, we would prefer to retain this term.
Round 2
Reviewer 2 Report
In the revised version the authors have made an effort to improve the paper and it has been enhanced considerably. However, I still feel that the figures are simply too cluttered and the time-scale on the x-axis should be coalesced in bi- or tri-monthly data for better visualization. There are still glitches in language, but to a much lesser degree. Suspect these will be corrected further down the process. At large, the paper is still extremely wordy and the text should be shortened.
Author Response
Dear Sir,
we have slightly shortened the manuscript further, and addressed some minor errors.
Furthermore, we reviewed the figures. Now, figure 1 and 3 are on three-months basis, and not monthly. Plus, we reorganized figure 2, and we hope it is now more easy to read.
We hope that these modifications will satisfy your requests.
Best regards,
on behalf of the authors,
Stefano Martellos